# The Benefice of Mobile Parts’ Exchange in the Management of Infected Total Joint Arthroplasties with Prosthesis Retention (DAIR Procedure)

**DOI:** 10.3390/jcm8020226

**Published:** 2019-02-09

**Authors:** Stefanie Hirsiger, Michael Betz, Dimitrios Stafylakis, Tobias Götschi, Daniel Lew, Ilker Uçkay

**Affiliations:** 1Orthopaedic Surgery Service, University of Geneva, Geneva 1211, Switzerland; Stefanie.Hirsiger@insel.ch (S.H.); Michael.Betz@balgrist.ch (M.B.); dimitrios.stafylakis@cuge.ch (D.S.); 2Balgrist University Hospital, Zurich 8008, Switzerland; tobias.goetschi@zurichmed.com; 3Service of Infectious Diseases, Geneva University Hospitals and Faculty of Medicine, Geneva 1211, Switzerland; daniel.lew@unige.ch

**Keywords:** antibiotic duration, DAIR, treatment failure, mobile parts’ exchange, orthopaedic surgery

## Abstract

***Background:*** The management of prosthetic joint infections (PJI) with debridement and retention of the implant (DAIR) has its rules. Some authors claim that lacking the exchange of mobile prosthetic parts is doomed to failure, while others regard it as optional. ***Methods:*** Single-center retrospective cohort in PJIs treated with DAIR. ***Results:*** We included 112 PJIs (69 total hip arthroplasties, 9 medullary hip prostheses, 41 total knee arthroplasties, and 1 total shoulder arthroplasty) in 112 patients (median age 75 years, 52 females (46%), 31 (28%) immune-suppressed) and performed a DAIR procedure in all cases—48 (43%) with exchange of mobile parts and 64 without. After a median follow-up of 3.3 years, 94 patients (84%) remained in remission. In multivariate Cox regression analysis, remission was unrelated to PJI localization, pathogens, number of surgical lavages, duration of total antibiotic treatment or intravenous therapy, choice of antibiotic agents, immune-suppression, or age. In contrast, the exchange of mobile parts was protective (hazard ratio 1.9; 95% confidence interval 1.2–2.9). ***Conclusions:*** In our retrospective single-center cohort, changing mobile parts of PJI during the DAIR approach almost doubled the probability for long-term remission.

## 1. Introduction

Prosthetic joint infection (PJI) management requires both surgery and antimicrobial therapy. The surgical options include one- or two-stage implant exchange, resection arthroplasty (with or without arthrodesis), or DAIR (debridement, antibiotics, irrigation, and retention). DAIR itself can have two goals: cure [1,2] or (life-long) suppressive therapy. For curative DAIR, literature is sparse regarding the number of surgical lavages needed [3] and the scientific proof for the recommended exchange of mobile parts (polyethylene, liners) [4]. These are important questions, because the consequences may lead to added expenses, prolongation of surgery (thus potentially enhancing surgical site infections [5]), and increased morbidity [4]. In this study, we epidemiologically determine the role of mobile parts’ exchange in the remission of DAIR.

## 2. Methods

Geneva University Hospitals are the only public hospital system in Geneva and some parts of neighboring France, with decades of experience in treating PJIs. In the DAIR approach, the exchange of mobile parts is recommended but finally left to surgeons’ decision and skills. 

We have performed a retrospective cohort study with our DAIR approaches since 2004. Most of the patients reported in this side study also participated in prospective studies [6,7,8,9] and in the Geneva Arthroplasty Register [10]. We reported the PJI definition in prior publications [1,11,12], which are based on the Infectious Diseases Society of America (IDSA) definitions [13]. A PJI required the presence of intraoperative pus and/or several positive intraoperative microbiological samplings. We considered remission as the absence of any clinical, laboratory, or imaging evidence of recurrence of the original infection. For this study, we excluded cases with incomplete documentation, minimal follow-up for less than 6 months, special pathogens such as mycobacteria or fungi, recurrent PJIs, surgery with complete explantation, or cases that we did not treat by ourselves. 

The indication for the DAIR approach accorded with international recommendations, e.g., the presence of relatively susceptible pathogens, the absence of a sinus tract, sufficient soft tissue coverage, the absence of implant loosening, and an acute infection [13]. We thus excluded chronic PJIs and those with sinus tracts or prosthesis loosening and performed DAIR in acute surgical site infections [5] or acute PJI’s in terms of late hematogenous seedings [1]. There were no chronic low-grade smoldering PJIs in our study population. DAIR, by itself and its definition, implies to keep the fixed parts (with its eventual prior cementation) in situ and to exchange only the mobile parts, if ever. We do not change the cement or introduce a new one during the DAIR procedure. The indication for the exchange of mobile parts (i.e., head, glenosphere, and polyethylene or metallic liner) depended on the surgeon, the IDSA recommendations [13], and the availability of these mobile parts (which usually differ from one commercial product to another). 

After discharge, the patients were followed-up in our hospital or by an Infectious Diseases expert in close collaboration (DL). We included the last patient on 10 October 2017 and continued the general follow-up until 31 December 2018. We also performed a literature review regarding mobile parts’ exchange in DAIR, by including all available reports of the last twenty years that provide original data with more than 10 own cases, and by excluding subsequent publications of the same database.

### Statistical Analyses

We performed retrospective group comparisons using the Pearson-χ^2^, the Wilcoxon rank-sum tests, and Kaplan–Meier curves for the entire study population, and repeated them for the subgroup of patients with at least two years of clinical active medical follow-up. Cox regression determined associations with remission. We introduced independent variables known to be associated with a poor DAIR prognosis into the multivariate analysis, except for surgical interventions and antibiotic treatment, which we automatically included into the final model. Likewise, we analyzed the number of surgical lavages and the duration of antibiotic therapy as continuous and as categorical variables. We included eight predictor variables per outcome and checked key variables for interaction. We used STATA software (9.0, STATA^™^, College Station, TX, USA). *P* values ≤ 0.05 (two-tailed) were significant.

## 3. Results

### 3.1. Patients and Pathogens

Among 154 DAIR episodes treated in our center, we excluded 42 for various reasons (Figure 1). We finally analyzed 112 PJIs (69 total hip arthroplasties, 9 medullary hip prostheses, 41 total knee arthroplasties (of which 3 were rotational), and 1 total shoulder arthroplasty) in 112 patients (median age 75 years, 52 females (46%), 31 (28%) immune-suppressed, and 23 (21%) bacteremic). Overall, 33 arthroplasties were revisions of prior non-infected surgeries. The immune-suppressions were as follows: diabetes mellitus (*n* = 13), alcoholism (*n* = 7), cancer (*n* =5), cirrhosis CHILD C (*n* = 2), medications (*n* = 2), or mixed causes (*n* = 2). The median time delay between arthroplasty (implantation) and infection was 4.3 months (range, 0.5–120 months). We detected 46 different microbiological PJI patterns. The most frequently identified pathogens were *Staphylococcus aureus* (*n* = 29; 8 methicillin-resistant), skin commensals (coagulase-negative staphylococci, corynebacteria, micrococci, *Propionibacterium acnes*; *n* = 32), streptococci (*n* = 22), Gram-negatives (*n* = 16; 3 *Pseudomonas aeruginosa*), enterococci (*n* = 6), and polymicrobial PJIs (*n* = 12). Six episodes were culture-negative.

### 3.2. Treatment

Overall, 108 DAIR cases (96%) were intended for cure. Suppressive DAIR was intended only in four cases. The median number of surgical interventions for infection was 2 (range, 0–6), among which there was an exchange of mobile parts in 48 cases (43%). The median duration of total antibiotic treatment was 3 months (range, 1.5–6 months), with a median of 12 first days intravenously (range, 0–49 days). We used 62 different antibiotic regimens with the five most frequent drugs being vancomycin (*n* = 47), quinolones (*n* = 55), clindamycin (*n* = 27), amoxicillin/clavulanate (*n* = 20), and rifampicin (*n* = 55). The antibiotics provoked important side effects in 36 patients (32%), of which nine required the immediate stop of the medication: diarrhea (*n* = 14), nausea (*n* = 8), skin rash (*n* = 4), acute renal insufficiency (*n* = 3), or various other events such as transient hepatitis or mycosis. These adverse events occurred early in the course with a median delay of three weeks from the start.

### 3.3. Outcomes

After a median follow-up of 3.3 years (range, 1.9–7.7 years), a total of 94 patients (94/112; 84%) remained in remission. Eighteen episodes (16%) witnessed septic failures occurring after a median delay of 0.9 years (range, 0.5–1.5 years) following the first infection. Among these 18 failures, 11 were true microbiological recurrences with the initial pathogens and 7 were new infections. We ignored non-infectious failures. In group comparison (Table 1), remission was unrelated to prosthesis localization, pathogens, number of surgical debridements, duration of total antibiotic treatment and of intravenous therapy, choice of antibiotic agents, immune-suppression, or age. The same proportions were witnessed when analyzing only the subgroup of patients with a minimum of two years of active clinical follow-up (Table 2). In this subgroup of 85 episodes, 13 (15%) witnessed failures. Remission was still unrelated to prosthesis localization, pathogens, number of surgical debridements, immune-suppression, or age. The multivariate analysis confirmed our group comparisons (Table 3). 

In contrast, the exchange of mobile prosthetic parts was statistically protective. We saw this effect when analyzing all 18 failures (hazard ratio 1.9, 95% confidence interval 1.2–2.9) or when regarding only microbiological cure rates (excluding the 7 failures with new pathogens) (hazard ratio 1.7, 95% CI 1.1–2.6). As expected, the Kaplan–Meier curve confirmed our significant findings only for the period of the first 400 days. Thereafter, the curves paralleled each other with almost no differences after 1000 days (Figure 2). We summarize our literature review (role of mobile parts’ exchange) in Table 4.

## 4. Discussion

According to our cohort, changing of mobile parts of PJI during the DAIR approach increases the probability of long-term remission, independently of if we had a follow-up of one year or beyond two years. This effect was seen in the multivariate analysis, whereas we failed to reveal it significantly in crude group comparisons with a huge case-mix. Surprisingly, despite worldwide expert recommendations and international consensus [4,19], a scientific evaluation regarding this topic is sparse and inconclusive. Some authors deny an influence on the final outcome [19,25,26,29,35,36], while others systematically demand it [4,14,20,21,27,31,32]. According to their own publications, however, their compliance in performing these exchanges oscillates around 50%, or they simply do not report them [2] (Table 3). In our study, the hazard ratio in terms of benefice of the mobile parts’ exchange was almost two, as it was equally witnessed in the multicenter study of Lora-Tamayo et al. [34]. Choi et al. computed a potential three-fold benefice when performing polyethylene exchanges in infected knee arthroplasties [37], while colleagues from Oxford report a four-fold increase in hip PJIs, especially when infections were early with shorter than 6 weeks’ incubation period [19]. Of note, our indications for the DAIR approach are based on internationally accepted recommendations [13].

We could not detect other variables associated with remission. The duration of antibiotic prescription did not influence long-term success. Several authors’ groups reported remissions with “short” prescriptions of 6 weeks in DAIR [1,2,14,16,18,28,38]. Others advocated for even less in selected cases with excellent evolutions, such as three [18] or four weeks [29,37], suggesting that the antibiotic duration *per se* is not determinant in DAIR when surgery has been adequately performed. This is important, because every third patient in our cohort reported adverse events during the antibiotic course [12].

Besides the fact that our single-center study is retrospective, it has other limitations. First, the proportion of “curative” DAIR cases was very high (96%). These patients are foreseen to stop antibiotics after several weeks of treatment and should not be confounded with (life-long) suppressive therapy *per se.* Hence, we cannot pronounce on this latter group due to paucity of cases. Second, in our cohort, the median number of surgical interventions was two. This is higher than most author groups advocate [2,15,26,27,28,31,33], reporting only one debridement in their DAIR approach (with or without exchange of mobile parts). Moojen et al. specifically investigated this question and concluded that a single debridement, with additional surgery on indication, appears to be at least as successful for DAIR than multiple debridements [3]. We found one publication advocating that 1 single debridement was not sufficient. This is, however, a study of 43 PJIs exclusively due to MRSA [25]. Third, no specific antibiotic regimens, e.g., the combination with rifampicin [34] or ciprofloxacin [35], or specific pathogens such as *S. aureus* [11,15,16,23,31] or *Pseudomonas aeruginosa* [27,32], were associated with altered remission rates. This can be related to the small sample size of the individual strata. For example, enterococcal DAIR might have a worse outcome according to clinical personal experience of many expert groups [39], which we, however, cannot prove with only six own cases. Fourth, we analyzed the mobile parts’ exchange as a dichotomous parameter (done versus not done within a “black box” approach). We ignored details regarding the intervention, its duration, and unreported complications. The experience of the surgical team and the individual operation course might play a role that we cannot include in our final analysis. Fifth, we focus on infection remission and not on functional outcome. The role of the exchange of mobile parts regarding immediate or long-term mechanical sequelae, embedded in a wide range of case-mix, is statistically difficult to determine [28] and probably only feasible within a prospective trial that is specially designed for that question. Sixth, the median follow-up time was 3.3 years, but the minimum was six months. In the literature of implant-related orthopedic infections the follow up time is usually one year, in terms of occurrence of infection [5], and it is usually 1–2 years after the treatment of PJI by the one- or two-change exchange [11]. However, concerning DAIR, the ideal active follow-up time is less clearly advocated in the literature, and persistence or recurrence of clinical infection (failure) is usually witnessed much earlier compared to exchange procedures [1]. We therefore conducted a side analysis with episodes harboring an active clinical follow-up of at least two years and found very similar results. Finally, although 18 different key variables such as pertinent demographics (sex, age, American Society of Anesthesiologist’s Score (ASA-Score)), material (type of prosthesis, revision), infection (pathogens, pain, bacteremia) and therapy (antibiotic durations, administrations and combinations, number and type of surgeries) have been assessed in the multivariable analyses, there are still others which we could not include to avoid “statistical model overfitting” such as the KLIC-Score (Kidney, Liver, Index surgery, Cemented prosthesis and C-reactive protein value) [30] or femoral neck fractures. Likewise, materials are important. We simply could not consider all materials individually in the pre-operative and post-operative settings (e.g., cement, different metals, which theoretically may influence the pre-operative risks and the post-operative outcomes).

In conclusion, changing of mobile parts during DAIR likely increases the chance of long-term remission (doubling the hazard ratio in our study). As usual, we call for caution in the interpretation of our retrospective study data. For example, some relevant information to help clinical decision-making may be lacking, and our survivor analysis mainly highlights the first-year impact of the DAIR approach. We personally consider this exchange imperative and certainly more important than any antibiotic-related or demographic parameters. Nevertheless, because of the controversy in published literature, a well-designed meta-analysis (or “Cochrane review”) still remains warranted. [40]

## Figures and Tables

**Figure 1 jcm-08-00226-f001:**
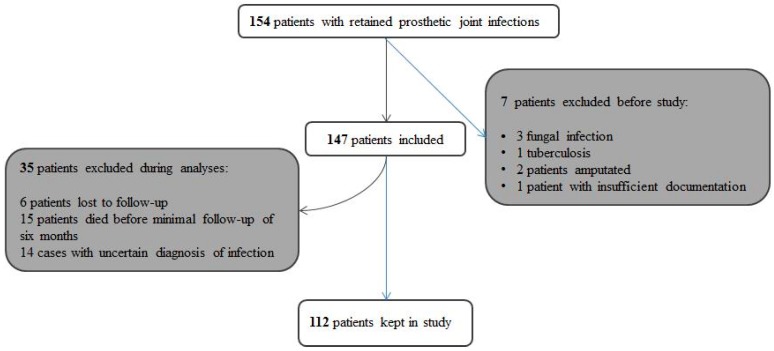
Flow chart displaying the inclusions and exclusions of our study.

**Figure 2 jcm-08-00226-f002:**
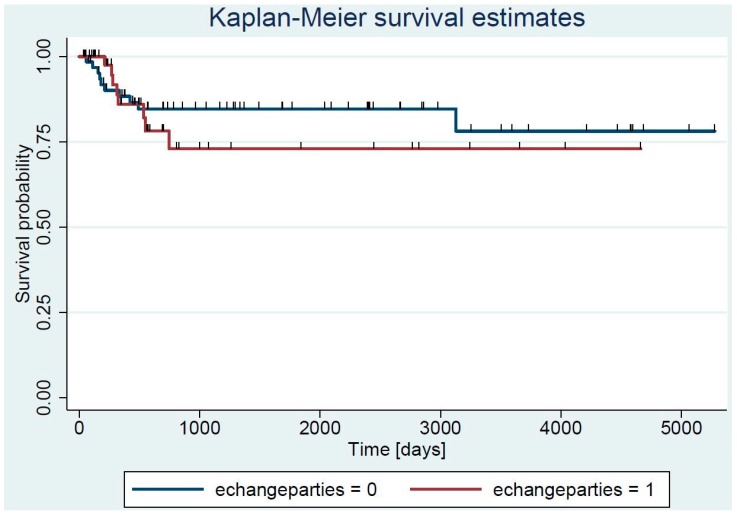
Kaplan–Meier curve stratified upon the exchange of mobile parts. Echangeparties = exchange of mobile parts. Of note, the benefit of exchange is only visible during the first year (first 400 days). Thereafter, the curves remain parallel. The corresponding hazard ratios and 95% confidence of the exchange are 1.9; 1.2–2.9; and, for non-exchange 0.5, 0.8–1.0, respectively.

**Table 1 jcm-08-00226-t001:** Comparison of demographic, clinical, and treatment characteristics of patients with the debridement and retention of the implant (DAIR) approach for prosthetic joint infections (PJI).

	Remission		Clinical Failure
Total *n* = 112	*n* = 94	*p* value	*n* = 18
Female sex	46 (49%)	0.224	6 (33%)
Age (median)	75 years	0.601	72 years
American Society of Anesthesiologists’ score (median)	3 points	0.991	3 points
Pain score on admission (median)	5 points	0.610	4 points
Total hip prostheses	58 (62%)	0.962	11 (61%)
Prior revision arthroplasty	26 (28%)	0.338	7 (39%)
Bacteremic infection	18 (19%)	0.406	5 (28%)
Infection due to *Staphylococcus aureus*	26 (28%)	0.329	3 (17%)
-Infection due to MRSA	8 (9%)	0.199	0 (0%)
-Infection due to streptococci	21 (22%)	0.101	1 (6%)
-Infection due to skin commensals	26 (28%)	0.625	6 (33%)
-Infection due to enterococci	4 (4%)	0.237	2 (11%)
-Infection due to gram-negative pathogens	11 (12%)	0.440	1 (6%)
Immune suppression ^+^	27 (29%)	0.572	4 (22%)
Number of surgical interventions (median)	2	0.973	2
-more than 1 intervention	49 (52%)	0.869	9 (50%)
-exchange of mobile parts	40 (43%)	0.882	8 (44%)
Duration of antibiotic treatment (median)	90 days	0.224	98 days
-100 days compared to ≤ 100 days	30 (32%)	0.304	8 (44%)
Duration of intravenous treatment (median)	10 days	0.416	14 days
-7 days compared to ≤ 7 days	61 (65%)	0.125	15 (83%)
Use of rifampicin-ciprofloxacin combination	46 (49%)	0.934	9 (50%)
Use of clindamycin	24 (26%)	0.420	3 (17%)
Use of amoxicillin/clavulanate	17 (18%)	0.886	3 (17%)
Use of vancomycin	38 (40%)	0.451	9 (50%)

^+^ Immune suppression = corticosteroid medication, organ transplantation, advanced cirrhosis, diabetes mellitus, alcoholism, or active cancer. MRSA = methicillin-resistant *Staphylococcus aureus;* DAIR = debridement, antibiotics, irrigation, and retention; PJI = prosthetic joint infection.

**Table 2 jcm-08-00226-t002:** Comparisons for the subset of 85 episodes with at least two years of active follow-up.

	Remission		Clinical Failure
Total *n* = 85	*n* = 72 (85%)	*p* value	*n* = 13 (15%)
Female sex	35 (49%)	0.500	5 (38%)
Age (median)	73 years	0.840	73 years
Immune suppression ^+^	25 (35%)	0.411	3 (23%)
Number of surgical interventions (median)	2	0.598	2
Exchange of mobile parts	26 (36%)	0.492	6 (46%)

^+^ Immune suppression = corticosteroid medication, organ transplantation, advanced cirrhosis, diabetes mellitus, alcoholism, or active cancer.

**Table 3 jcm-08-00226-t003:** Univariate and multivariate analyses of factors potentially related to remission of retained infected arthroplasties (results expressed as hazard ratios with 95% confidence intervals).

Total *n*= 112	Univariate Analysis	Multivariate Analysis
Female sex	0.8, 0.5–1.3	n.d.
Age (median)	1.0, 1.0–1.0	n.d.
American Society of Anesthesiologists’ score (median)	0.9, 0.7–1,2	n.d.
-ASA score 2 compared to 1	1.0, 0.8–1.3	n.d.
-ASA score 3 compared to 1	1.4, 0.5–4.0	n.d.
-ASA score 4 compared to 1	1.2, 0.4–3.5	n.d.
Pain score on admission (median)	1.1, 0.3–3.5	n.d.
Total hip prostheses	0.8, 0.5–1.2	n.d.
Revision arthroplasty	0.8, 0.5–1.3	n.d.
Bacteremic infection	1.3, 0.7–2.1	n.d.
Infection due to *Staphylococcus aureus*	1.0, 0.6–1.5	n.d.
-Infection due to MRSA	1.0, 0.5–2.1	1.1, 0.5–2.3
Infection due to streptococci	1.3, 0.8–2.1	n.d.
Infection due to enterococci	1.2, 0.4–3.4	n.d.
Number of surgical interventions (median)	0.8, 0.6–1.0	0.7, 0.5–1.1
-more than 1 intervention	0.7, 0.5–1.1	n.d.
-exchange of mobile parts	***2.0, 1.3–3.0***	***1.9, 1.2–2.9***
Duration of antibiotic treatment (median)	1.0, 1.0–1.0	1.0, 1.0–1.0
-100 days compared to ≤ 100 days	1.6, 0.9–2.5	n.d.
Duration of intravenous treatment (median)	1.0, 1.0–1.0	n.d.
-7 days compared to ≤ 7 days	0.8, 0.5–1.2	n.d.
Use of rifampicin-ciprofloxacin combination	1.2, 0.8–1.8	n.d.

*** Significant values ≤ 0.05 (two-tailed) are displayed ***in bold and italic***. MRSA = methicillin-resistant *Staphylococcus aureus,* ASA = American Society of Anesthesiologists’ score, n.d. = not done.

**Table 4 jcm-08-00226-t004:** Prosthetic joint infections treated with DAIR—selected articles with at least 10 own cases; published since 1997.

Author	Number PJI	Main Pathogen	Identified Key Variables for Success	Exchange Mobile Parts	Remission Incidence and Remarks
Mont et al. [14]	24 knees	*S. aureus*	Early PJI	24 (100%)	83%
Marculescu et al. [15]	99	*S. aureus*	Early PJI (< 8 days), absence fistula	48 (48%)	46%
Deirmengian et al. [16]	31	*S. aureus*	Lack of *S. aureus*	10 (32%)	35%, exchange no benefice
Theis et al. [17]	73	*S. aureus*	Early PJI (< 4 weeks)	not reported	69%
Tsumura et al. [18]	10	*S. aureus*	none	0 (0%)	80%
Grammatopoulos [19]	122 hips	*S. aureus*	Mobile parts’ exchange, (< 6 weeks)	65 (53%)	68%, four-fold benefice of exchanging
Buller et al. [20]	309	Gram-positives	Early PJI (< 3 weeks)	309 (100%)	52%
Gardner et al. [21]	44 knees	*S. aureus*	none	44 (100%)	43%
Vilchez et al. [22]	53	*S. aureus*	Serum CRP < 22 mg/L, 1 debridement	not reported	76%
Koyonos et al. [23]	138	*S. aureus*	*S. aureus*	not reported	35%
Puhto et al. [24]	113	staphylococci	Leukocyte count < 10 G/L	not reported	62%
Peel et al. [25]	43	MRSA	>1 debridement, antibiotics <3 months	18 (42%)	86%, exchange no benefice
Achermann et al. [26]	50	staphylococci	Early PJI (< 3 weeks)	26 (52%)	92%, exchange no benefice
Sukeik et al. [27]	26 hips	staphylococci	Early PJI (< 5 days)	26 (100%)	77%
Westberg et al. [28]	38	*S. aureus*	Serum CRP < 10 mg/L	not reported	71%
Geurts et al. [29]	89	*S. aureus*	Early PJI (< 4 weeks)	0 (0%)	83%
Kuiper et al. [30]	91	staphylococci	Coagulase-negative staphylococci	not reported	66%
Fehring et al. [17]	86	*S. aureus*	none	not reported	37%
Moojen et al. [3]	68 hips	*S. aureus*	none	not reported	79%
Konigsberg et al. [31]	42	staphylococci	Lack of *S. aureus*	42 (100%)	76%
Duque et al. [32]	67	*S. aureus*	not MRSA and not *P. aeruginosa*	67 (100%)	69%
Sendi et al. [33]	30 hips	staphylococci	none	14 (47%)	90%
Lora-Tamayo et al. [34]	444	streptococci	Mobile parts’ exchange	220 (50%)	58%, two-fold benefice of exchanging
Chaussade et al. [1]	87	*S. aureus*	Lack of MRSA	87 (100%)	69%
Rodriguez-Pardo [35]	174	Gram-negatives	Ciprofloxacin treatment	96 (55%)	68%, exchange no benefice
Choi et al. [36]	28 hips	*S. aureus*	Lack of *S. aureus*	19 (68%)	50%, exchange no benefice
Choi et al. [37]	32 knees	*S. aureus*	Mobile parts’ exchange	19 (59%)	31%, three-fold benefice of exchanging
*Present study*	*112*	*S. aureus*	*Mobile parts’ exchange*	*48 (43%)*	*84%, two-fold benefice of exchanging*

DAIR = debridement, antibiotics, irrigation, and retention; MRSA = methicillin-resistant *S. aureus;* ASA = American Society of Anesthesiologists’ score; PJI = prosthetic joint infection; CRP, C-reactive protein.

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
