# Peer review of "The Benefice of Mobile Parts’ Exchange in the Management of Infected Total Joint Arthroplasties with Prosthesis Retention (DAIR Procedure)"

_jcm, 2019, doi:10.3390/jcm8020226_

Reviewer 1 Report

 The authors address a topic of some interest - not completely resolved- as is the role of the removal of mobile parts in the surgery of PJI with retention of the implant (DAIR). Although the importance of this action in this type of management is currently assumed, this has not been reliably demonstrated. The authors conclude through their study that the removal of the moving parts is associated with a higher probability of healing.

Some aspects should be outlined:

the authors point out that they use their own definition of PJI; it would be more appropriate to use a standardized definition (MSIS, IDSA)

the prognosis of the patients is globally good, so I suppose that the patients had an adequate indication for DAIR, although this should be clarified in the text

the minimum follow-up to establish the clinical cure should be at least 1 year, and ideally 2 years; only patients who meet these criteria should be included

in the multivariate analysis variables with p<0.05 are included; it is recommended the inclusion of all the variables that in previous studies have been associated with a poor DAIR prognosis, as well as those that in the multivariate analysis have a p <0.2 or 0.3

it is striking that, in the univariate analysis, the percentage of patients with mobile parts replacement is practically the same in the group with clinical failure than in the group with clinical cure (which makes it difficult to understand the final results of the multivariate analysis); do you have any explanation?

it is not clear what is the contribution of the review of studies that do not evaluate the role of removal of mobile parts to the conclusions of the authors

Author Response

Reviewer: 1

The authors address a topic of some interest - not completely resolved- as is the role of the removal of mobile parts in the surgery of PJI with retention of the implant (DAIR). Although the importance of this action in this type of management is currently assumed, this has not been reliably demonstrated. The authors conclude through their study that the removal of the moving parts is associated with a higher probability of healing.

Answer: Yes. This is exact.

 Some aspects should be outlined:

 The authors point out that they use their own definition of PJI; it would be more appropriate to use a standardized definition (MSIS, IDSA)

Answer: We only referenced prior articles instead of repeating in detail the PJI definitions, which all in fine base on the IDSA guidelines. Now we reference the IDSA guidelines also in this manuscript; number 13 (page 2, lines 49-50).

 The prognosis of the patients is globally good, so I suppose that the patients had an adequate indication for DAIR, although this should be clarified in the text

Answer: Yes, we say and define now in the Methods section, also by citing the IDSA guidelines (page 2, lines 55-66). And we also repeat it in the Discussion (page 8, lines 167-168).

 The minimum follow-up to establish the clinical cure should be at least 1 year, and ideally 2 years; only patients who meet these criteria should be included

Answer: Yes and no. In detail, the follow-up time is one year in terms of infection risk after an implant surgery (CDC consensus). It is usually 1-2 years after the treatment of PJI by one or two-change exchange. However, in DAIR the ideal active follow-up time is less clear. The persistence or clinical infection (failure) is usually witnessed much earlier compared to the exchange procedures.

 However, we understand the Reviewer and propose a supplementary analysis specifically for the subgroup of patients with more than 2 years of active follow-up, instead of deleting the information (bias) for the whole study population. There is no difference in this subgroup analysis.

 We explain now all this in the Statistics section (page 2, lines 75-76), Results (page 3, lines 118-114), a new Table 2 (page 5), and the Limitation (page 8, lines 204-205). Meanwhile, within the submission process, the follow-time has also increased for the majority of the patients.

 In the multivariate analysis variables with p<0.05 are included; it is recommended the inclusion of all the variables that in previous studies have been associated with a poor DAIR prognosis, as well as those that in the multivariate analysis have a p <0.2 or 0.3.

Answer: Ok, we reword (page 2, lines 77-78).  This doesn’t change the message at all.

 It is striking that, in the univariate analysis, the percentage of patients with mobile parts replacement is practically the same in the group with clinical failure than in the group with clinical cure (which makes it difficult to understand the final results of the multivariate analysis); do you have any explanation?

Answer: We re-run the analysis and re-confirmed our results. It remains as it stays. Usually, crude group comparisons and multivariate analyses are not the same. Even if the tendency may be seen in both for good reason, most scientific publications display both of them, because there might be differences between both statistical analyses. Personally, we rely rather on the multivariate analysis. We mention now this discrepancy briefly in the Discussion (page 7, lines 157-158).

 It is not clear what is the contribution of the review of studies that do not evaluate the role of removal of mobile parts to the conclusions of the authors.

Answer: It is the e sake of completeness of our literature review and our personal choice. We certainly can delete them if the Editor insists, but we prefer to keep them in order to display  all literature at a glance. By this, the audience can also judge what is currently present or absent in the literature.

Reviewer 2 Report

Abstract: you need to state clearly the surgical procedure you performed. Were the modular parts exchanged? how many patients were involved? 

inclusion criteria not acceptable. Chronic infections? acute infections? the shoulder arthroplasty should be excluded: include only hip and knee

Revisions should not be compared to primary surgeries. The septic risk is very different.

Explain what you mean with mobile parts. Why don't you remove the fixed components? any loosening components? What were the criteria you used to decide when a fixed part haad to be left in situ? And what about mobile parts?

a minimum 2 year follow-up is required when you evaluate PJI 

details about implants are lacking (cement? antibiotic cement?)

The idea is potentially interesting, but requires to be analyzed in details. 

Author Response

Reviewer: 2

Abstract: you need to state clearly the surgical procedure you performed.

Answer: Right. We mention it now in the Abstract, page 1, line 22.

 Abstract: Were the modular parts exchanged?  How many patients were involved?

Answer: Yes. We display it now in the Abstract, page 1, lines 22-23.

 Inclusion criteria not acceptable.

Answer: We beg to differ. Now, we explicitly reference the IDSA guidelines; number 13 (page 2, lines 49-50). These are all international definitions, on which we only add our DAIR cases. DAIRitself is defined per acronym (DAIR = debridement, antibiotics, irrigation, and retention). Our inclusion criteria are usual and nothing is particular.

 Chronic infections?

Answer: Reviewer 2 is right. We have to mention this explicitly that there are no chronic infections in DAIR (page 2, line 60).

 Acute infections?

Answer: Yes, they are all acute postoperatively, or acute in terms of late hematogenous PJI There were no chronic low-grade smoldering PJIs in our study population (page 2, lines 59-60).

 The shoulder arthroplasty should be excluded: include only hip and knee.

Answer: This would be a bias, as we intend not to exclude any DAIR procedure. Other Reviewers do not want to exclude the single shoulder infections we have. The message does not change and is the same.

 Revisions should not be compared to primary surgeries. The septic risk is very different.

Answer: In our study, we have only infected patients which undergo the DAIR procedure, which nota bene in itself is a revision. Reviewer 2 might eventually mix up prevention of infection with the outcome of infected arthroplasties?

 Certainly Reviewer 2 is right by saying that revision surgeries harbor more infection risk than primary arthroplasties. However, if an arthroplasty is infected, the literature does not say that there is more microbiological failure risk depending on the number of prior uninfected arthroplasties in a given patient. Past uninfected surgeries (e.g. 10 years ago) might rather influence mechanical outcomes, but not the outcome of infection treatment. Indeed, in our study, a prior revision failed to influence the success of the later DAIR (Table 1, Table 3).

 Explain what you mean with mobile parts.

Answer: Right. We give now some examples on page 2, line 64.

 Why don't you remove the fixed components?

Answer: Of course we have removed the fixed components for all PJI cases that we treated with a one- or two-stage exchange. However, this exchange population with prosthesis removal is not part of this study and thus is not mentioned here. Our manuscript only concerns DAIR cases, which are mostly retained (especially the fixed parts) by definition (DAIR = debridement, antibiotics, irrigation, and retention).

 Any loosening components?

Answer: No prior loosening in DAIR. We say it now on page 2, line 59.

 What were the criteria you used to decide when a fixed part had to be left in situ?

Answer: DAIR, by itself and by definition, implies to keep the fixed parts in situ and to exchange only the mobile parts, if ever. We explicitly repeat it now on page 2, lines 61-62.

 And what about mobile parts?

Answer: On page 2, lines 62-66.

 A minimum 2 year follow-up is required when you evaluate PJI

Answer: In detail, the follow-up time is one year in terms of infection risk after an implant surgery (CDC consensus). It is usually 1-2 years after the treatment of PJI by one or two-change exchange. However, in DAIR the ideal active follow-up time is less clear. The persistence or clinical infection (failure) is usually witnessed much earlier compared to the exchange procedures.

 However, we understand the Reviewer and propose a supplementary analysis specifically for the subgroup of patients with more than 2 years of active follow-up, instead of deleting the information (bias) for the whole study population. There is no difference in this subgroup analysis.

 We explain now all this in the Statistics section (page 2, lines 75-76), Results (page 3, lines 118-114), a new Table 2 (page 5), and the Limitation (page 8, lines 204-205). Meanwhile, within the submission process, the follow-time has also increased for the majority of the patients.

 Details about implants are lacking (cement? antibiotic cement?)

Answer: The question of antibiotic-containing cements primarily regards prevention aspects in terms of occurrence of PJI, or therapeutic aspects in the one-stage exchange or the two-stage exchange (i.e. antibiotic-containing spacers).

 In the DAIR procedure, the implant (and its cement) is retained by definition. Therefore there is no changing of cement; and there is also no new cementing. It is rather the cement of the index arthroplasty that is usually maintained, but this occurs far before the occurrence of any infection.  We now explicitly state that we do not change the cement or introduce a new one during the DAIR procedure (page 2, line 63).

Round  2

Reviewer 1 Report

The authors have made a commendable effort and have substantially improved the manuscript. However, questionable aspects remain. Among the most relevant is the lack of adjustment for variables that have been associated with a higher frequency of DAIR failure, such as the KLIC-score. It is true that the results of the univariate analysis can change significantly when adjusting for other variables in the multivariate analysis. However, it draws the attention that, if we look at the group of patients with 2 years of follow-up, there is even a higher percentage of mobile parts exchange in the group with clinical failure; it would be necessary to seek an explanation for the change of direction between the result of the univariate and multivariate analysis (for example, in which group of patients with clinical failure this exchange was more often performed? the potential confounding variable should be identified in order to interpret the results). It is true that the duration of follow-up necessary to establish the cure of PJI is not well established (in none of the surgical procedures); however, in a two-stage replacement, intraoperative cultures may be a good criterion of cure, which is not available in the case of DAIR (so in the latter case, the prolonged clinical follow-up is of greater importance and a follow-up of at least one year is generally accepted).

Author Response

The authors have made a commendable effort and have substantially improved the manuscript. However, questionable aspects remain. Among the most relevant is the lack of adjustment for variables that have been associated with a higher frequency of DAIR failure, such as the KLIC-score.

Answer: Statistically speaking you cannot overstretch a multivariate model and you must remain with a maximum of independent variables per outcome parameter. There are statistical rules that limit the incorporation of a high number of variables, which is also called “model overfitting”. In our multivariate analysis (Table 3), we already include key parameters y such as pertinent demographics (sex, age, ASA-Score), material (type of prosthesis, revision), infection (pathogens, pain, bacteremia) and treatment (antibiotic duration, administration and combinations, number and type of surgeries). Overall, we already compute and display 18 items and cannot incorporate deliberately.

 Regarding the KLIC-Score, Reviewer 1 argues that this is a key variable in DAIR and should be implemented in all publications, which we obviously cannot do retrospectively. KLIC-Score is new and probably Reviewer 1 is its main promoter in the scientific community. A recent PubMed search on 22 January 2019 displays 80 articles with “DAIR”; and only 3 of them with “KLIC-Score”.

 However, we now acknowledge that we cannot include more variables in the multivariate analysis and that we lack the KLIC-Score, which we reference (new reference 36, page 8, lines 205-210) with one of the three articles found in PubMed.

 It is true that the results of the univariate analysis can change significantly when adjusting for other variables in the multivariate analysis. However, it draws the attention that, if we look at the group of patients with 2 years of follow-up, there is even a higher percentage of mobile parts exchange in the group with clinical failure; it would be necessary to seek an explanation for the change of direction between the result of the univariate and multivariate analysis (for example, in which group of patients with clinical failure this exchange was more often performed? the potential confounding variable should be identified in order to interpret the results).

Answer: We see no major confounding.

In Table 1. the proportions with a mobile parts exchange are displayed as

    -     exchange of mobile parts

40 (43%)

0.882

8 (44%)

In Table 2 with the subgroup of patients with a minimum of two-years follow-up, the proportions with a mobile parts exchange are displayed as

-       exchange of mobile parts                   26 (36%)              0.492                                 6 (46%)

The percentages of exchange are around 40% in all groups and the small differences are also possible “just by chance”. Of note, the differences are statistically neutral and by far insignificant.

 It is true that the duration of follow-up necessary to establish the cure of PJI is not well established (in none of the surgical procedures); however, in a two-stage replacement, intraoperative cultures may be a good criterion of cure, which is not available in the case of DAIR (so in the latter case, the prolonged clinical follow-up is of greater importance and a follow-up of at least one year is generally accepted).

Answer: We have now a median follow-up of 3.3 years and performed a separate analysis (Table 2) with a minimum follow-up time of 2 years. Hence we show that the results did not change according to general and long follow-up times, which represent additional information. Moreover, we acknowledged the different follow-up times as a limitation (page 8, lines 198-203).

Reviewer 2 Report

Now better. I still think that mixing hip with knee surgeries is not a good idea and it is a trick to raise numbers (a very diffuse trick). Mixing revisions with primary surgeries is definitely a bad idea: you know what revision surgery means and it is simply not comparable to primary implants. In my opinion, the lack of consensus about PJI is mainly due to mixed populations and confusing inclusion criteria. Moreover, the use of very short follow-up is not acceptable and may contribute to confusion. The progression of knowledge should start from these points. Available literature about infections is too frequently poor, lacks systematic approaches and promotes generalizations. This article does the same.  

The materials are important. You simply did not consider them in the pre-operative and post-operative settings (I mean couplings, cement, metals, which may influence the pre-operative risks and the post-operative outcomes). I think that it is a significant flaw. However, this is not the focus of your study, and statistics would not support another evaluation. 

However, you wrote an article which is in line with the current literature. Now the article is acceptable, but to me, has a little merit due to the very important flaws related to inclusion criteria, surgical techniques, follow-ups etc. I will leave it to readers' judgements.

Author Response

Reviewer: 2

Now much better. I still think that mixing hip with knee surgeries is not a good idea and it is a trick to raise numbers (a very diffuse trick).

Answer: There is no “trick” behind every study protocol. The argumentation of selling an article goes in both directions. For example, we could also “trick” in terms of doubling the numbers of publications by splitting the same analysis for every arthroplasty separately (e.g. knee, hip, and with and without cement; with this and that metal surface, or other materials); which we, as a matter of fact did not do so. It is usual in the literature of orthopedic infections to analysis hip and knee arthoplasties together (avoiding selection biases). Our paper is no exception and rather the rule.

 Mixing revisions with primary surgeries is definitely a bad idea: you know what revision surgery means and it is simply not comparable to primary implants. In my opinion, the lack of consensus about PJI is mainly due to mixed populations and confusing inclusion criteria.

Answer: Reviewer 2 addresses general problems, which we agree partially. However, we perform a study in line with the literature and avoid biases. Moreover, our multivariate analysis (Table 3) already contains a variable “revision”.

 Moreover, the use of very short follow-up is not acceptable and may contribute to confusion.

Answer: We have now a median follow-up of 3.3 years and performed a separate analysis (Table 2) with a minimum follow-up time of 2 years. Hence we show that the results did not change between general and long follow-up times, which represent additional information. Moreover, we fully acknowledged the existence of different follow-up times as a limitation (page 8, lines 198-203).

 The progression of knowledge should start from these points. Available literature about infections is too frequently poor, lacks systematic approaches and promotes generalizations. This article does the same. 

Answer: We understand that Reviewer 2 obviously has a lot of personal experience in publishing papers in this particular field of research. Of course we cannot argue against his/her general statements, but general shortcomings  are not the “fault” of our research group, or of this specific paper.

 The materials are important. You simply did not consider them in the pre-operative and post-operative settings (I mean couplings, cement, metals, which may influence the pre-operative risks and the post-operative outcomes). I think that it is a significant flaw. However, this is not the focus of your study, and statistics would not support another evaluation. 

Answer: Yes. However, the argument and the wording are good and we now explicitly acknowledge it as an additional limitation of the study (page 8, 210-212).

 You wrote an article which is in line with the current literature. Now the article is acceptable, but to me, has a little merit due to the very important flaws related to inclusion criteria, surgical techniques, follow-ups etc. I will leave it to readers' judgements.

Answer: Thank you. Reviewer 2 understands that the paper is written in line with the literature. General shortcomings of retrospective studies or of osteoarticular studies are beyond the possibility of a single research group and definitively need international consensus in the scientific community. Likewise, we cannot not stratify and purify the study population for single materials, single patients groups, homogenous materials, and homogenous approaches and so on.

 This said: if Reviewer 2 agrees, our research group would be delighted to collaborate with him/her on a future and brief opinion paper about general shortcomings in this particular academic field of research.